# Dual-Band Antenna with Pattern and Polarization Diversity

**DOI:** 10.3390/s24155008

**Published:** 2024-08-02

**Authors:** Jungmin Mo, Youngje Sung

**Affiliations:** Department of Electronic Engineering, Kyonggi University, Suwon 16227, Republic of Korea; mojung0725@kyonggi.ac.kr

**Keywords:** high isolation, pattern–diversity antenna, polarization diversity

## Abstract

This study proposes a pattern–diversity antenna with different radiation patterns at two different frequency bands (*f*_1_ and *f*_2_; *f*_1_: broadside radiation pattern, *f*_2_: conical radiation pattern). The proposed structure consists of a central circular antenna and two annular ring antennas, with each of the antennas having individual ports. Two of the ports (port 1 and port 3) exhibit orthogonal broadside radiation patterns at low bands, and the other two ports (port 1 and port 2) exhibit orthogonal conical radiation patterns at high bands. Thus, they have polarization diversity characteristics. To improve isolation between the ports, the inner part of the annular ring antennas is shorted by an array, and the outermost port is positioned orthogonal to the other ports. Using this configuration, the isolation values between the ports are −26.7 and −30.1 dB at the two frequency bands, respectively. Using the fabricated prototype, experimental results show that the proposed antenna achieves −10 dB bandwidths of 240 MHz (5.71–5.95 GHz) and 210 MHz (7.69–7.9 GHz) at *f*_1_ and *f*_2_, respectively.

## 1. Introduction

Dual-band antennas enable the simultaneous use of two communication links with a single antenna. Additionally, when applied to GPS systems, dual-band technology can provide improved accuracy compared with the use of a single frequency. By incorporating additional frequency bands, dual-band antennas can enhance data quality and connection success rates, addressing the limitations of using a single frequency band [1].

Dual-band antennas can be implemented using two methods. The first involves positioning two or more independent antennas close together to create a dual-band antenna [2,3], and the second involves the use of a single antenna [4,5,6,7]. The former is becoming less common because of the relatively large size. Dual-band antennas with a single resonator use substrate-integrated waveguide (SIW) technology [4,5] or diplexing methods [6]. These methods share the same structure and can be seamlessly integrated into a common design. However, the frequency ratio is small.

A radiation pattern indicates the directionality and strength of the transmitted signal. Antennas have unique radiation patterns, which are selected based on the usage requirements. Radiation pattern diversity is a key technology in systems such as multiple-input–multiple-output (MIMO) antennas. The use of different radiation patterns can increase data transmission rates, mitigate interference, and minimize the impact of multipath effects.

Many studies have implemented radiation-pattern diversity. A single port can be used to obtain different radiation patterns at multiple frequency bands. This has been achieved using antennas based on artificial magnetic conductors [7] or by applying spiral inverted-F antennas [8]. Also, a circular slot can be loaded in the circular patch to enhance the impedance matching at the higher frequency band, thus widening the operating bandwidth [9]. Multiple ports at specific frequencies can be used to achieve different radiation patterns. Methods that arrange ports vertically [10] or use annular ring antennas with shorted inner parts have been used to achieve isolation between ports [11]. Furthermore, a combination of inductively loaded patches and annular ring antennas has been employed to diversify radiation patterns and polarization [12,13]. Also, rectangular and V-like-shaped slits are used for isolation enhancement and antenna miniaturization [14]. Multiple ports at different frequency bands can be used to obtain different radiation patterns. This can be achieved by aligning the centers of circular antennas and annular ring antennas [15]. In another method, a sequentially rotated microstrip feed is used for the high band and four open-ended slots are used for an omni-directional pattern in the low band [16]. Shielded mushroom structures in artificial transmission lines have been used to implement dual-band triple-mode antennas [17].

In this study, we achieved dual-band characteristics using a circular antenna and an annular ring antenna with its inner part shorted by an array. We obtained two different radiation patterns at two frequency bands and simultaneously obtained identical but orthogonal radiation patterns at the same frequency band. The presented metrics are simulation results obtained using HFSS software. The measured data were obtained through experiments conducted using the fabricated prototypes.

## 2. Antenna Configuration

Figure 1 shows the configuration of the proposed antenna. The substrate used in the design is shown in orange with RO4003C, with a thickness of 1.58 mm, dielectric constant of 3.55, and loss tangent of 0.0027. The proposed antenna comprises one circular antenna and two annular ring antennas, shorted by an array and is shown in blue. The radii of the vias located inside are 0.27 mm, and those of the vias located outside are 0.6 mm. The outermost ring antenna (Antenna I) has outer and inner radii denoted as *r*_1_ and *r*_2_, respectively. Antenna II has outer and inner radii denoted as *r*_3_ and *r*_4_, respectively. A circular antenna with a radius of *r*_5_ (Antenna III) is positioned in the center. The ground radius is set to *r*_0_. The positions of the ports are determined by *d*_1_, *d*_2_, and *d*_3_.

Antennas II and III are positioned inside the ring structure of Antenna I, enabling the antennas to achieve different radiation patterns (namely, broadside and conical radiation patterns) at two different frequency bands without increasing the size of the antenna. To enhance isolation between ports, vias are incorporated into the structure. Antenna I has 30 vias and Antenna II has 60 vias. Antennas I and III form a broadside radiation pattern at 5.93 GHz in the TM_11_ mode. Antennas I and II form a conical radiation pattern at 7.92 GHz in the TM_41_ and TM_21_ modes, respectively. Port 1 is positioned orthogonal to Ports 2 and 3 for good isolation. Furthermore, the vias located on the outer side improve the isolation characteristics between the antennas.

The resonant frequencies of the TM modes of the proposed antennas can be determined using (1) [18,19].
(1)$$f_{nm}=\frac{X_{nm}c}{2\pi a_{e}\sqrt{\varepsilon_{r}}}$$

In this equation, nm is related to the TM_nm_ mode, c represents the speed of light, and $\varepsilon_{r}$ denotes the dielectric constant of the substrate. $a_{e}$ denotes the effective radius, which can be obtained using (2). $X_{nm}$ is the root of (3), and its values are $X_{11}$ = 1.8412, $X_{21}$ = 1.3406, and $X_{41}$ = 2.5876.

In (2), a represents the radius of a circular antenna and the inner radius of a ring antenna; h denotes the height of the substrate. Equation (3) is derived from the boundary conditions of the cavity mode for these antennas. Here, g is the ratio of the outer radius to the inner radius, and $J_{n}$ and $N_{n}$ denote the nth order Bessel functions of the first and second kind, respectively, where the prime notation indicates the first derivative.
(2)$$a_{e}=a\sqrt{1+\frac{2h}{\pi a\varepsilon_{r}}\left(ln\frac{\pi a}{2h}+1.776\right)}$$
(3)$$J_{nm}^{\prime }\left(k_{nm}\right)N_{n}\left(k_{nm}g\right)-J_{n}\left(k_{nm}g\right)N_{nm}^{\prime }\left(k_{nm}\right)=0\;$$

## 3. Parameter Study

### 3.1. Isolation Based on Port Positions

Figure 2a shows the antenna configurations corresponding to different port positions. Antenna I shows three ports aligned in a straight line. Antenna II shows a 90° rotation of port 2. Antenna III shows a 90° rotation of port 1. Antenna IV shows a 90° rotation of ports 1 and 2. The antenna parameters, except for the port positions, are set as follows: The antenna dimensions are $r_{0}=25\;\mathrm{mm},\;r_{1}=21.5\;\mathrm{mm},\;r_{2}=15\;\mathrm{mm},\;r_{3}=13.97\;\mathrm{mm},\;r_{4}=8.75\;\mathrm{mm},\;\mathrm{and}\;r_{5}=7.45\;\mathrm{mm}$. The radius of the via for Antenna 1 is 0.6 mm, and for Antenna II, it is 0.27 mm. The positions of the three ports in the proposed structure are set as follows: $d_{1}=18.4\;\mathrm{mm},\;d_{2}=10.78\;\mathrm{mm},\;\mathrm{and}\;d_{3}=2.3\;\mathrm{mm}$. Figure 2b shows the simulation results for isolation as the antenna ports are rotated by 90°. Figure 2c,d show the simulation results for isolation at frequencies of 5.93 and 7.92 GHz, respectively. From the simulation results, when ports 1 and 3 are perpendicular, the isolation characteristics are good.

As shown in Figure 2c, the isolation between ports 1 and 3 of each model at the 5.9 GHz band is −13 dB (Antennas I and II, where the two ports are horizontal). In comparison, Antennas III and IV, where the two ports are perpendicular, show an improved isolation of approximately −30 dB. Port 2 does not significantly influence the isolation characteristics based on its position relative to the other two ports. As shown in Figure 2d, comparing the S_21_ at 7.92 GHz, all three antenna models exhibit isolation values of approximately −30 dB. Based on these results, Antenna III was selected to enhance the isolation characteristics between the antenna ports.

Good isolation can also be confirmed using the E-field of each antenna. The regions close to red indicate strong electric fields, and those close to blue indicate weak fields. Figure 3a shows the E-field simulation result confirmed when the electric power is supplied to port 1 at 5.93 GHz. For Antennas I and II, the electric field is also applied to port 3, and thus isolation is not well performed. However, for Antennas III and IV, the electric field becomes null on the horizontal axis where port 3 is located, affecting isolation between ports. In addition, Figure 3c shows the E-field simulation result, confirmed when the electrical power is supplied to port 3 at 5.93 GHz. For Antennas I and II, the electric field is also applied to port 1, and thus isolation is not well performed. By contrast, for Antennas III and IV, the electric field is null on the horizontal axis where port 3 is located, affecting isolation between the ports. Figure 3b shows the E-field simulation result confirmed when the electric power is supplied to port 1 at 7.92 GHz, and Figure 3d shows the E-field simulation result confirmed when the electric power is supplied to port 2 at 7.92 GHz. In both cases, all four antenna models do not affect other ports and have excellent isolation characteristics. Based on these results, Antenna III is selected to enhance the isolation characteristics between the antenna ports.

### 3.2. Effect of Number of Vias

According to [20], adjusting the number of vias in the antenna can control the resonant frequencies corresponding to each TM mode and adjust the bandwidth or separate the bands.

Figure 4a shows the simulation results with different reflection coefficients based on the number of vias located on Antenna I to demonstrate the change in the angle between the vias. The angles between the vias were set to 5°, 12°, and 15°. All other antenna parameters were kept constant. As shown in Figure 4, no significant difference in performance based on the number of vias in S_33_ was observed. However, for S_11_, as the angle between the vias increased (with few vias), the resonant frequency decreased, as indicated by the green diamonds in Figure 4a. With an angle of 12°, a wider bandwidth existed below −10 dB than with angles of 5° and 15°. For S_22_, a 12° angle provided the widest bandwidth, below −10 dB. Based on these results, the angle between the vias was set to 12°.

### 3.3. Effect of Antenna Size

Figure 5 shows the simulation results for reflection coefficients and isolation characteristics with different lengths *r*_1_. The length of *r*_1_ was increased from 21 mm in 0.5 mm increments, and all other antenna parameters were kept constant. As shown in Figure 5a, no significant performance difference in S_22_ with changes in the size of *r*_1_ was observed. However, for S_11_, as *r*_1_ increased (as the size of Antenna I increased), the resonant frequency decreased, as indicated by the green diamonds in Figure 5a. For S_33_, no bandwidth below −10 dB with a size of 21 mm was observed. With a size of 22 mm, the bandwidth was significantly narrower than with a size of 21.5 mm. Based on these results, the radius *r*_1_ of Antenna I was set to 21.5 mm.

Figure 6 shows the simulation results for reflection coefficients and isolation characteristics with different values of *r*_3_. The length of *r*_3_ was increased from 13.8 mm in 0.1 mm increments, and all other antenna parameters were kept constant. As shown in Figure 6a, *r*_3_ does not have a significant effect on S_11_ and S_33_. However, for S_22_, as *r*_3_ increases (as the size of Antenna II increases), the resonant frequency decreases, as indicated by the green diamonds in Figure 6a. Based on these results, the radius *r*_3_ was set to 13.97 mm and the isolation values for each operating band were less than −20 dB.

Figure 7 shows the simulation results for reflection coefficients and isolation characteristics with different values of *r*_5_. *r*_5_ was increased from 7.25 mm in 0.2 mm increments, and all other antenna parameters were kept constant. As shown in Figure 7a, no significant performance difference in S_22_ with changes in the size of *r*_5_ was observed. Although variations in the resonant frequency and reflection coefficient values were observed for S_11_, they were not significant. However, for S_33_, as *r*_5_ increased, the resonant frequency decreased, as indicated by the green diamonds in Figure 7a. Additionally, the bandwidth below −10 dB was widest when the size was 7.45 mm. Based on these results, the radius *r*_3_ was set to 13.9 mm. At this size, the isolation values for each operating band were less than 20 dB. 

## 4. Simulated and Measured Results

Figure 8 shows a photograph of the proposed antenna, fabricated using the optimized parameters. Figure 9 compares the simulated and measured reflection coefficients and isolation characteristics of the proposed antenna. The bandwidth of −10 dB for the two frequency bands is shown in Figure 9a. For *f*_1_, the common bandwidth between ports 1 and 3 is within the range of 5.71–5.95 GHz, resulting in a bandwidth of 210 MHz. Figure 9b shows the isolation characteristics for the ports at each resonant frequency. The isolation between ports 1 and 2 is approximately −35.5 dB. The measured results also confirm good isolation between the ports. The resonant frequencies were shifted approximately 150 MHz lower than the simulated results. This discrepancy between simulation and measurement results with a slight shift was due to the extra length of the fabricated antenna.

Figure 10 shows the electrical field formed on the antenna surface as radio-frequency (RF) power is applied to different ports at resonant frequencies. The regions close to red indicate strong electric fields, and those close to blue indicate weak fields. Figure 10a shows the simulation results for the E-field when RF power is applied to port 1 at 5.93 GHz. The electrical field along the horizontal axis where port 3 is located becomes null, indicating that the ports do not influence each other. Similarly, Figure 10b shows the simulation results when RF power is applied to port 3 at 5.93 GHz. The electrical field along the vertical axis where port 1 is located becomes null, indicating no influence between the ports. As shown in Figure 10c,d, the electrical field is extremely weak at ports other than the one where RF power is applied, indicating no influence. The simulation results show that the proposed antenna exhibits excellent isolation characteristics.

Figure 11 compares the measured radiation patterns of the proposed antenna with the simulated results. As shown in Figure 11a,b, at 5.93 GHz, broadside patterns are formed at ports 1 and 3, with energy radiating in the +z-axis direction. At 7.92 GHz, conical patterns are formed at ports 1 and 2, as shown in Figure 12a,b, respectively. The measured results closely match the simulated values, indicating similar characteristics. The measured antenna gains are approximately 8.2 dBi for port 1 and 8.8 dBi for port 3 at 5.93 GHz. At 7.92 GHz, the gains are approximately 4.9 dBi for port 1 and 6.9 dBi for port 2.

## 5. Conclusions

This study proposed an antenna with both pattern and polarization diversity. We implemented a pattern–diversity antenna with different radiation patterns in two different frequency bands. The proposed antenna comprised one circular antenna and two ring antennas, with the inner section shorted by an array. Each antenna used a feeding method using probes. To improve isolation between the ports, each port was designed to be orthogonal. As a result, the isolation values between ports were −26.7 and −30.1 dB at the two frequency bands, respectively. Based on its excellent isolation characteristics, the proposed antenna exhibited broadside radiation pattern characteristics at *f*_1_ and conical radiation pattern characteristics at *f*_2_, perpendicular to each other. Thus, the proposed antenna is expected to show great potential for multi-band mobile communication applications. Table 1 presents the comparison of the performance of the proposed antenna with previously published dual-band microstrip antennas.

## Figures and Tables

**Figure 1 sensors-24-05008-f001:**
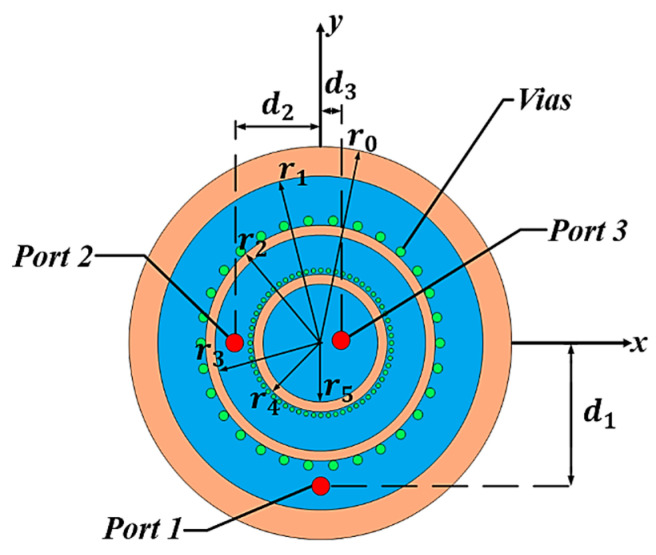
Geometry of the proposed antenna.

**Figure 2 sensors-24-05008-f002:**
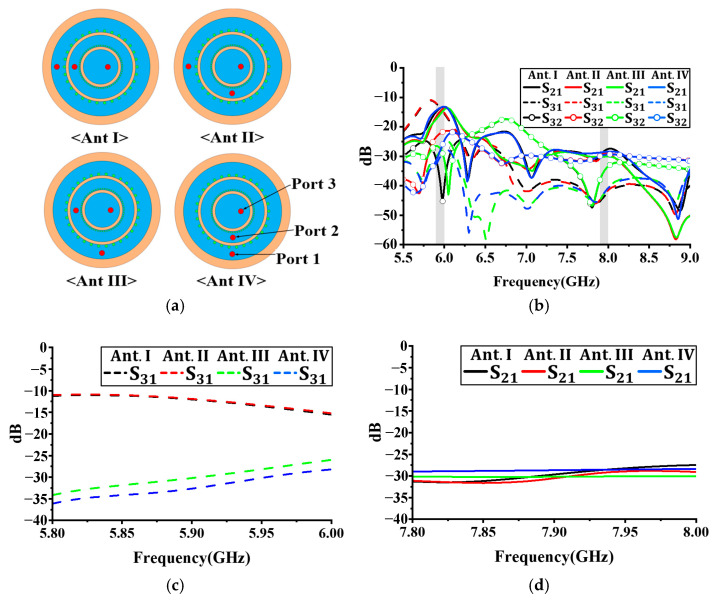
Simulation results of port positions rotated by 90°: (**a**) proposed antenna structure; (**b**) isolation; (**c**) isolation at lower band; (**d**) isolation at higher band.

**Figure 3 sensors-24-05008-f003:**
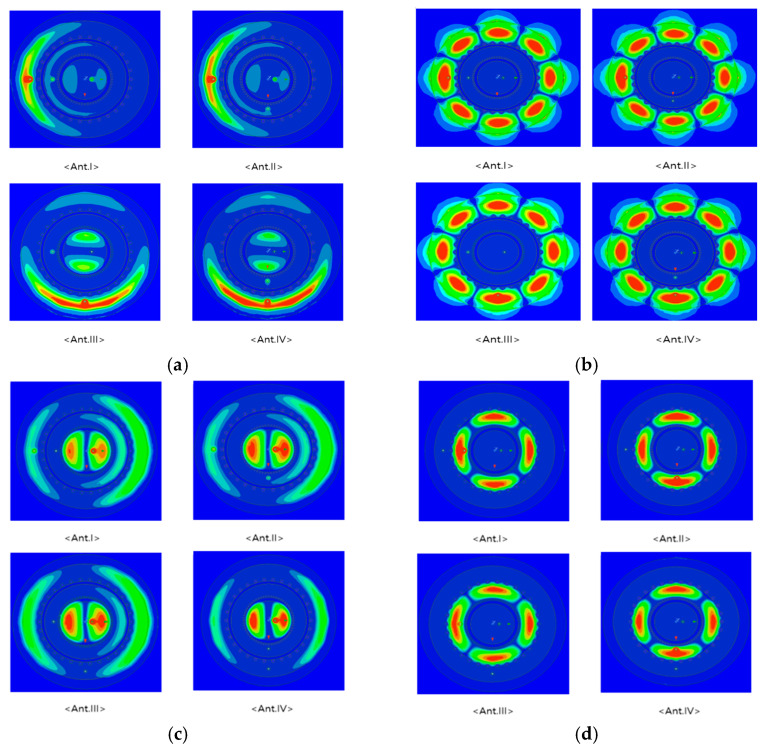
E-field distribution: (**a**) at 5.93 GHz_port 1; (**b**) at 7.92 GHz_port 1; (**c**) at 5.93 GHz_port3; (**d**) at 7.92 GHz_port2.

**Figure 4 sensors-24-05008-f004:**
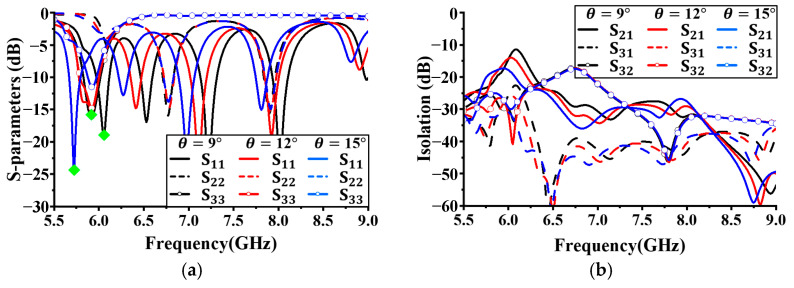
Simulated antenna performance with different angels (θ) between the vias: (**a**) reflection; coefficient; (**b**) isolation.

**Figure 5 sensors-24-05008-f005:**
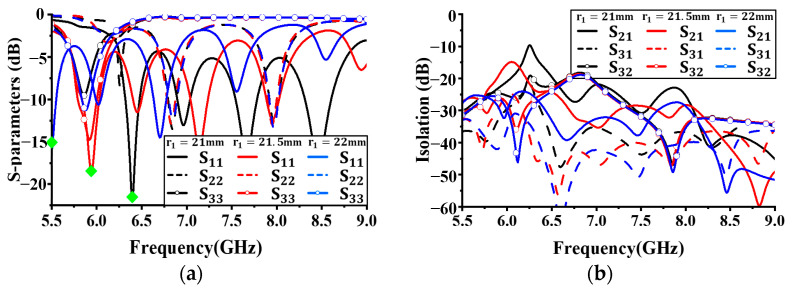
Simulated antenna performance with different radius *r*_1_: (**a**) reflection coefficient; (**b**) isolation.

**Figure 6 sensors-24-05008-f006:**
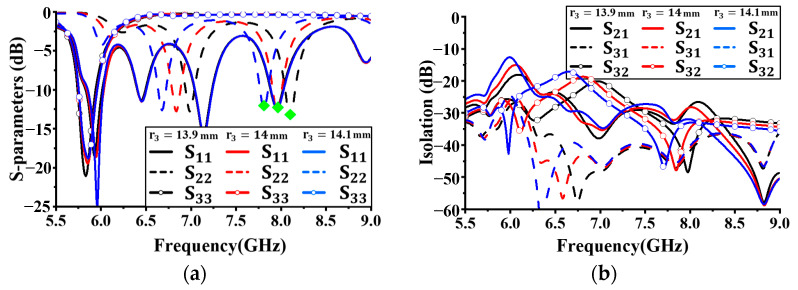
Simulated antenna performance with different radius *r_3_*: (**a**) reflection coefficient; (**b**) isolation.

**Figure 7 sensors-24-05008-f007:**
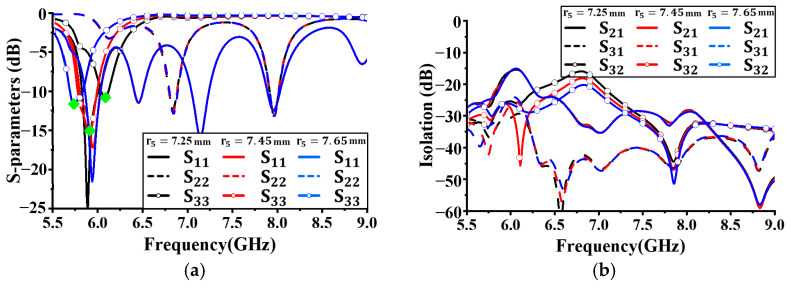
Simulated antenna performance with different radius *r_5_*: (**a**) reflection coefficient; (**b**) isolation.

**Figure 8 sensors-24-05008-f008:**
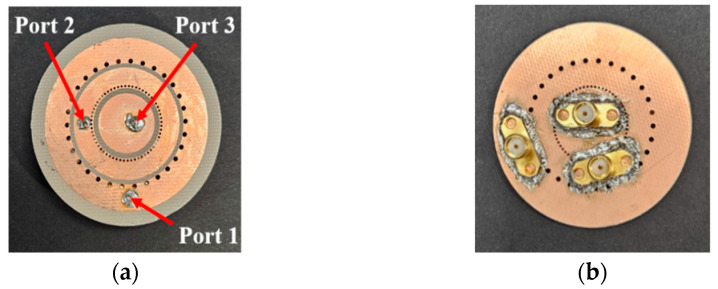
Prototype of the proposed antenna: (**a**) top layer; (**b**) bottom layer.

**Figure 9 sensors-24-05008-f009:**
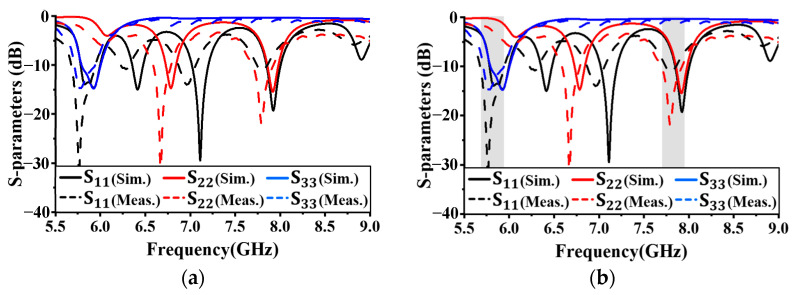
Simulated and measured results: (**a**) reflection coefficient; (**b**) isolation.

**Figure 10 sensors-24-05008-f010:**
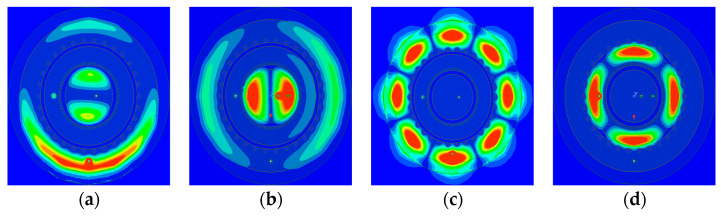
Simulated E-field distribution: (**a**) 5.93 GHz @ port 1; (**b**) 5.93 GHz @ port 3; (**c**) 7.92 GHz @ port 1; (**d**) 7.92 GHz @ port 2.

**Figure 11 sensors-24-05008-f011:**
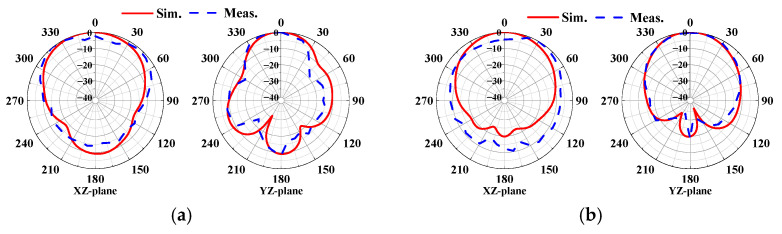
Normalized radiation pattern at 5.93 GHz: (**a**) port 1; (**b**) port 3.

**Figure 12 sensors-24-05008-f012:**
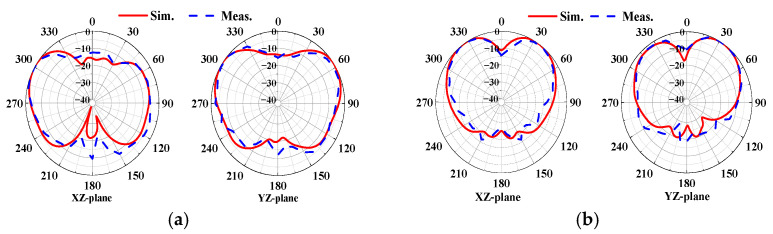
Normalized radiation pattern at 7.92 GHz: (**a**) port 1; (**b**) port 2.

**Table 1 sensors-24-05008-t001:** Comparison with previous dual-feed dual-cp antennas.

Ref.	Freq. [GHz]	BW [%]	Size	# of Port	Iso. [dB]	Pattern
[2]	5.8/30	3.1/8.1 17/6.3	1.90$\lambda_{0}\;\times$ 1.90$\lambda_{0}\;\times$ 0.020$\lambda_{0}$	2	<−25	O/B
[4]	5.2/5.8	6/3.4	1.50$\lambda_{0}\;\times$ 1.30$\lambda_{0}\;\times$ 0.038$\lambda_{0}$	2	−28	B/B
[5]	8.3/10.5	1.9/2.7	0.88$\lambda_{0}\;\times$ 0.72$\lambda_{0}\;\times$ 0.040$\lambda_{0}$	2	−27.9	B/B
[6]	2.5/5.3	4.5/5.5	0.57$\lambda_{0}\;\times$ 0.57$\lambda_{0}\;\times$ 0.010$\lambda_{0}$	2	−21	B/B
[7]	1.4/1.6	2/1	0.57$\lambda_{0}\;\times$ 0.57$\lambda_{0}\;\times$ 0.030$\lambda_{0}$	1	X	O/B
[8]	2.5/5.8	4/3.4	0.30$\lambda_{0}\;\times$ 0.30$\lambda_{0}\;\times$ 0.080$\lambda_{0}$	1	X	O/B
[10]	3.5	1.4	0.35$\lambda_{0}\;\times$ 0.35$\lambda_{0}\;\times$ 0.035$\lambda_{0}$	3	−13/−13/−25	O/B/B
[11]	5.8	2.5	0.96$\lambda_{0}\;\times$ 0.96$\lambda_{0}\;\times$ 0.030$\lambda_{0}$	2	−26.5	O/B
[14]	3.75	1.1	0.29$\lambda_{0}\;\times$ 0.29$\lambda_{0}\;\times$ 0.015$\lambda_{0}$	4	<−10	O/B
[15]	3.31/5.86	3.9/6.2	1.68$\lambda_{0}\;\times$ 1.33$\lambda_{0}\;\times$ 0.44$\lambda_{0}$	3	<−25	B/O
[16]	2.5/5.9	8.1/10.3	0.40$\lambda_{0}\;\times$ 0.40$\lambda_{0}\;\times$ 0.035$\lambda_{0}$	2	−14/−16	O/B
This work	5.9/7.9	3.6/3.6 1.5/2.1	0.49$\lambda_{0}\;\times$ 0.49$\lambda_{0}\;\times$ 0.030$\lambda_{0}$	3	−26.7/−30.1	B/O

## Data Availability

The data that support the findings of this study are available from the corresponding author upon reasonable request.

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
