# Peer review of "Dual-Band Antenna with Pattern and Polarization Diversity"

_sensors, 2024, doi:10.3390/s24155008_

Round 1
Reviewer 1 Report
Comments and Suggestions for Authors
An antenna with both pattern and polarization diversity is proposed in this manuscript. However, there are several problems.
1)“ Two of the ports exhibit orthogonal broadside radiation 10 patterns; the other two ports exhibit orthogonal conical radiation patterns.”
How many ports are there in this antenna design
2)The format shown in Figure 4 is incorrect, and there is also Figure 9
3)In Figures 10 and 11, simulation data cannot be seen
4)“Isolation based on port positions”, Why not use isolation instead of using an electric field distribution diagram that is not intuitive
5)The references in the study lack relevant research from recent years, and most of the comparative literature is from 5 years ago.
Comments on the Quality of English LanguageMinor editing of English language required
Reviewer 2 Report
Comments and Suggestions for Authors
This work deals with the design of dual-band antenna with pattern and polarization diversity. Author had proposed a pattern-diversity antenna with different radiation patterns at two different frequency bands.
Author has to explain how he has designed and obtained the final proposed dimensions of the proposed antenna.
In the same time there is a difference between simulation and measurement results for reflection and isolation coefficients. Author has to explain why this difference.
For figure 10 and 11 author should plot the both results simulation and measurement permitting to compare them.
At the end author has to insert a Table comparing the proposed study with other studies published in literature.
Comments on the Quality of English Language
Minor editing of English language required
